# pET28g: A Golden Gate-compatible pET vector for protein expression in *Escherichia coli*, validated by production of functional human ACE2

Inês M. Luís, Mariana Parada, João B. Vicente, Isabel A. Abreu◉*

Instituto de Tecnologia Química e Biológica António Xavier, Universidade Nova de Lisboa (ITQB NOVA), Oeiras, Portugal

* abreu@itqb.unl.pt

## Abstract

The pET28g plasmid is a new tool for protein expression in *Escherichia coli*. It was derived from pET28a by replacing the traditional multiple cloning site with a Golden Gate cassette containing the *lacZα* reporter gene. The assembly of pET28g is based on Golden Gate cloning, and it was designed as an extension of the MoClo Toolkit (Addgene Kit #1000000044). The pET28g plasmid, along with the level 0 modules developed in this work, expands the existing pET plasmid collection and offers a versatile and modular system for protein expression. These tools facilitate the establishment of efficient pipelines for the preparation and testing of multiple constructs, enabling the optimization of conditions to achieve high levels of soluble protein expression. As proof of concept, we successfully produced and purified the peptidase domain of human ACE2 using the pET28g system. Notably, this represents the first report of the successful purification of a functional human ACE2 protein from the *Escherichia coli* soluble protein fraction. This lab protocol, which is complemented by the protocol at www.protocols.io (https://www.protocols.io/view/preparation-of-level-0-modules-for-golden-gate-ass-j8nlk9e75v5r/v1), provides a comprehensive guide to understanding the nomenclature of the pET28g system and includes step-by-step instructions for the preparation of level 0 modules and the assembly of the final plasmid to express constructs of interest using pET28g.

## Introduction

The pET expression plasmid was developed by Studier and colleagues in the late 1980s by incorporating the T7 RNA polymerase φ10 promoter (commonly referred to as the T7 promoter) and the T7 transcription terminator into a pBR322 backbone [1], thereby creating a plasmid system specifically designed for expression using T7 RNA polymerase [2]. This work laid the groundwork for the widely adopted pET series, a

**Data availability statement:** All relevant data are within the manuscript and its Supporting Information files.

**Funding:** This publication received funding from Fundação para a Ciência e a Tecnologia (Portugal) for Fellowship for IML (PD/BD/113982/2015) and grants FilliGrain-Protect (DOI:10.54499/PTDC/ASP-PLA/1920/2021), Green-it Base (DOI:10.54499/UIDB/04551/2020) and Programmatic (DOI:10.54499/UIDP/04551/2020) Funding, LS4FUTURE Associated Laboratory (DOI:10.54499/LA/P/0087/2020) all for IAA; MostMicro Programmatic Funding (UIDP/04612/2020), and iNOVA4Health R&D Unit (UIDB/04462/2020, UIDP/04462/2020) for JBV. Work was also funded by "la Caixa" Foundation and FCT, I.P. under the project code LCF/PR/HP22/52320025 (HR22-00722) to IAA. The funders had no role in study design, data collection and analysis, decision to publish, or preparation of the manuscript.

**Competing interests:** The authors have declared that no competing interests exist.

system that enables high-level protein expression in *Escherichia coli* (*E. coli*) strains containing the DE3 lysogen, which encodes the T7 RNA polymerase. Since then, the pET series has expanded to include a wide variety of expression plasmids, supporting efficient transcription and becoming a standard tool for protein production in many research applications. Over 100 different pET-based plasmids are now available and include various tags for affinity purification, antibody recognition, or improved solubility, and different selection markers. Today, several pET toolboxes are available in biological repositories or through commercial suppliers. Additionally, several optimizations have been made to improve *E. coli* strains and culture media to solve issues like leaky expression, codon bias, or protein toxicity (as summarized in [3,4]).

Golden Gate cloning, described in 2008 by Engler, Kandzia, and Marillonnet, is a highly efficient DNA assembly method that uses type IIS restriction enzymes, which cut outside their recognition sequences [5]. This enables seamless assembly of multiple DNA fragments into a destination plasmid in a single reaction, with the desired construct being selectively retained through iterative cycles of restriction and ligation. The use of type IIS sites allows for the generation of multiple distinct fusion sites with a single enzyme, facilitating the in-tandem assembly of large and complex constructs. This makes it particularly useful in synthetic biology for building transcriptional units and multi-gene pathways [6,7]. This versatile technology provides a fast, building block-based system for assembling and testing transcription units for protein expression.

pET plasmids are well-known for their robust protein expression capabilities; however, a collection of distinct plasmids and different cloning strategies is often necessary to test different transcription units with varied tags, protease recognition sites, or signal peptides. In the Golden Gate cloning strategy, the range of possibilities for assembling different transcription units is defined by the extent of the modules in the library [6]. These modules are, by definition, level 0 plasmids that contain multiple distinct elements, which can be combined and assembled in tandem into a destination plasmid, referred to as the level 1 plasmid. By converting the pET28a plasmid into a Golden Gate level 1 plasmid, the pET system has the potential to become a modular and flexible platform that integrates the Golden Gate cloning technology.

Here, we report the pET28g plasmid, a modified version of pET28a, in which the traditional multiple cloning site has been replaced with a Golden Gate cassette featuring *lacZα* selection. Designed as an extension of the MoClo Toolkit (Addgene Kit #1000000044) [6,8], pET28g simplifies the assembly of protein expression constructs. Additionally, we have developed modules (level 0 plasmids) that enable the assembly of protein expression constructs with different, cleavable, N-terminal protein tags by using the acceptor plasmids provided with the MoClo Toolkit (Addgene Kit #1000000044) to build new level 0 modules. These modules serve as a foundation for a plasmid collection that offers a streamlined and flexible solution for protein expression. A detailed protocol on how to prepare level 0 modules and assemble them into the level 1 pET28g plasmid is available at www.protocols.io under the https://doi.org/10.17504/protocols.io.j8nlk9e75v5r/v1, and can be accessed through the link: https://www.protocols.io/view/preparation-of-level-0-modules-for-golden-gate-ass-j8nlk9e75v5r/v1. Alternatively, the deposited protocol can be found in the S1 Protocol available in the Supporting Information.

## Materials and methods

The MoClo plasmids referenced throughout this work are part of the MoClo Toolkit deposited in Addgene with the catalogue number 1000000044 [6,8].

### Cloning of a Golden Gate cassette into pET28a

The Golden Gate cassette containing a *lacZα* reporter gene was amplified from the MoClo plasmid pICH47761 using the following pair of primers: 5'-CAATCCCATGGAGACCGCAGCTGGCACG (forward) and 5'-ATGACTCGAGTCAGCGTGAGACCGTCACAGCT (reverse). The amplification product was digested with the NcoI and XhoI enzymes and ligated into the backbone of pET28a (Novagen), previously digested with the same restriction enzymes. The final plasmid was transformed into *Escherichia coli* DH5α cells and selected in Luria Bertani (LB)-agar plates containing 50 mg/L kanamycin, 100 mg/L 5-Bromo-4-chloro-3-indolyl β-D-galactopyranoside (X-Gal), and 100 μM Isopropyl β-D-1-thiogalactopyranoside (IPTG). The final plasmid extracted from positive colonies is the pET28g, which consists of a pET28a backbone with a Golden Gate cassette inserted between the ribosome binding site (RBS) and T7 terminator, replacing the original Multiple Cloning Site (MCS). The Golden Gate cassette inserted is flanked by the fusion sites CCAT and CGCT, denominated fusion sites (FS) 1 and 7 for systematization (Table 2), which can be exposed by digestion with the BsaI enzyme (Fig 1).

### Preparation of N-terminal tag modules

Modules containing N-terminal tags were cloned between fusion sites 1 and 3 (Table 2). The MoClo Toolkit universal acceptor plasmid, pAGM9121, was used to prepare these modules. The maltose binding protein (MBP) tag was amplified from the pMAL-c2X (New England Biolabs) upon domestication of the MBP coding sequence to introduce a silent mutation that eliminates the BsaI recognition site. The tag containing 8×His fused with immunoglobulin G-binding domain 1 (Gb1) of Streptococcal Protein G was amplified from a plasmid containing the DNA sequence encoding for 8×His-Gb1 protein. The primers used for amplification are described in Table 1 and include the BpiI recognition site (italicized in the primer sequences), the adapter for the universal acceptor plasmid (underlined in the primer sequences), and the fusion sites 1 (CCAT) and 3 (AGGT) (highlighted in bold in the primer sequences) (see Fig 2B2 for a schematic representation of the primer design). The amplification product was ligated to the backbone of the MoClo pAGM9121 plasmid using Golden Gate cloning technology with BpiI enzyme. The final plasmids were transformed into *E. coli* DH5α cells and selected LB-agar plates containing 100 mg/L spectinomycin, 100 mg/L X-Gal, and 100 μM IPTG. The plasmids extracted from positive colonies are level 0 modules used for level 1 assembly in pET28g and were named p028g13-MBP and p028g13-8His_Gb1.

### Preparation of protease cleavage site modules

The module containing the cleavage site for Tobacco Etch Virus (TEV) protease was prepared by annealing oligonucleotides with complementary sequences and distinct flanking regions (see Fig 3B3 for a schematic representation of the oligonucleotide design). The sequences of the oligonucleotides annealed to clone TEV cleavage site (TEVs) were 5'-**AGGT**GAAAACCTGTATTTTCAGGG (F1_GG_TEVs_AGGT) and 5'-**CGAA**CCCTGAAAATACAGGTTTTC (R1_GG_TEVs_TTCG). These oligonucleotides are complementary in the region that codes for the peptide sequence ENLYFQG and are flanked by either fusion sites 3 (AGGT) or 4 (TTCG) (highlighted in bold in the oligonucleotide sequence). The product of the oligonucleotide annealing has the fusion sites exposed and can be directly used in the Golden Gate reaction using the MoClo acceptor plasmid pAGM1299 and BpiI enzyme. The final plasmid was transformed into *E. coli* DH5α cells and selected in LB-agar plates containing 100 mg/L spectinomycin, 100 mg/L X-Gal, and 100 μM IPTG. The final plasmid extracted from positive colonies, a level 0 module used for level 1 assembly in pET28g, was named p028g34-TEVs.

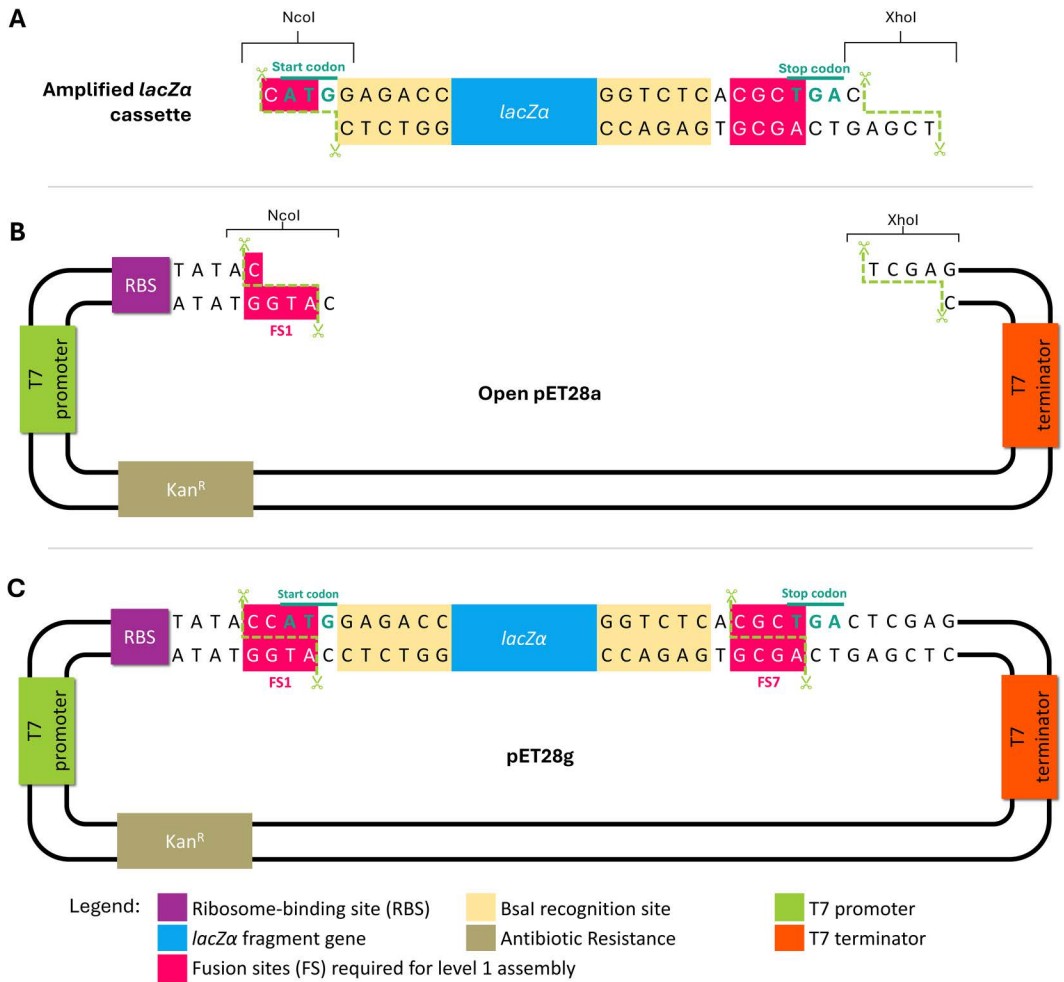

**Fig 1. Schematic representation of pET28g assembly. (A)** The Golden Gate cassette was amplified by PCR and digested with NcoI and XhoI enzymes, exposing the sticky ends that match those exposed in the (B) pET28a digested with the same pair of restriction enzymes. **(C)** The pET28g was obtained through ligation of the digested DNA and contains a lacZα reporter gene flanked by BsaI recognition sites that expose the sticky ends of fusion sites 1 (FS1) and 7 (FS7) (Table 2) used for level 1 assembly.

**Table 1. Primers used to construct the N-terminal tags.**

| Primer name | 5' to 3' sequence |
| --- | --- |
| F1_GG_MBP_CCAT | TT*GAAGACA*T<u>CTCA</u>**CCAT**GAAAATCGAAGAAGGTAAACTG |
| R1_GG_MBP_AGGT | TT*GAAGACA*A<u>CTCG</u>**ACCT**GACCCGAGGTTGTTGTTATTG |
| F1_GG_8His-Gb1_CCAT | TT*GAAGACA*T<u>CTCA</u>**CCAT**GGGCAGCAGCCATC |
| R1_GG_8His-Gb1_AGGT | TT*GAAGACA*A<u>CTCG</u>**ACCT**GATTCCGTAACCGTGAAGGTTTTGG |

## Cloning of the gene of interest

The gene of interest used to test this new plasmid encodes a truncated version of the human protein angiotensin-converting enzyme 2 (ACE2, UniProt Acc. No. Q9BYF1). The gene was amplified from a previously cloned pDONR221 plasmid containing the peptidase domain of the ACE2 gene (residues 19–615). Briefly, the gene with codon optimization

**Table 2. Fusion sites used in the MoClo Toolkit (Addgene, kit # 1000000044) and the numbers attributed for their identification on the pET28g cloning system.**

| Fusion site number | Fusion sites nucleotide sequence (5' to 3') |
|---|---|
| 1 | CCAT |
| 2 | AATG |
| 3 | AGGT |
| 4 | TTCG |
| 5 | GCTT |
| 6 | GGTA |
| 7 | CGCT |

for protein expression in *E. coli* was synthesized with attB adaptors and cloned into the pDONR221 plasmid (Thermo Fisher Scientific) using Gateway cloning. Then, the peptidase domain of ACE2 was amplified from the pDONR211 cloned with the ACE2 gene using the primers with the following sequences: 5'-TT*GAAGAC*AT<u>CTCA</u>**TTCG**ACCATTGAAGAA CAGGCGA (F1_GG_Ec-hACE2_TTCG) and 5'-TT*GAAGAC*AA<u>CTCG</u>**AGCG**TTAATCTGCATACGGGCTC (R1_GG_ Ec-hACE2_CGCT). The amplification product obtained with these primers contains the coding sequence for the truncated ACE2 flanked by fusion sites 4 (TTCG) and 7 (CGCT) (highlighted in bold in the primer sequences), the adapter for the universal acceptor plasmid (underlined in the primer sequences), and the BpiI recognition site to expose the adapter (italicized in the primer sequences). The amplification product was ligated to the backbone of the MoClo pAGM9121 plasmid using Golden Gate cloning technology with BpiI enzyme. The final plasmid was transformed into *E. coli* DH5α cells and selected in LB-agar plates containing 100 mg/L spectinomycin, 100 mg/L X-Gal, and 100 µM IPTG. The plasmid extracted from positive colonies, a level 0 module used for level 1 assembly in pET28g, was named p028g47-ACE2.

## Preparation of ACE2 expression plasmids – Level 1 assembly and pET28a cloning

Golden Gate ligations were carried out using the BsaI enzyme and combining the TEVs and ACE2 modules with either the MBP or the 8×His-Gb1 modules. The final plasmids were transformed into *E. coli* DH5α cells and selected in LB-agar plates containing 50 mg/L kanamycin, 100 mg/L X-Gal, and 100 µM IPTG. The plasmids extracted from positive colonies, level 1 plasmids with a complete transcription unit, were named pET28g-MBP_ACE2 or pET28g-Gb1_ACE2, depending on the N-terminal module used for level 1 assembly – for a detailed protocol, refer to pET28g protocol deposited at www.protocols.io (10.17504/protocols.io.j8nlk9e75v5r/v1, https://www.protocols.io/view/ preparation-of-level-0-modules-for-golden-gate-ass-j8nlk9e75v5r/v1).

As a control, we cloned the same truncated human ACE2 protein into pET28a. The restriction enzymes NheI and EcoRI were used both to digest the pET28a empty plasmid and to isolate the gene of interest from the pDONR221 plasmid containing the peptidase domain of ACE2, which was described above for the cloning of the level 0 module containing the gene. Both restriction products were purified and ligated to generate the plasmid pET28a-ACE2, with a thrombin-cleavable 6×His tag and a T7 tag at the N-terminus.

## Protein expression using pET28g constructs with two distinct N-terminal tags for added protein solubility

The pET28g and pET28a plasmids containing the cloned ACE2 gene were transformed into *E. coli* BL21 (DE3) cells for small-scale expression tests. Transformed cells were grown at 37 °C in LB broth supplemented with 50 mg/L kanamycin until they reached an optical density at 600 nm ($OD_{600}$) of 0.8–1.0. Protein expression was induced with 100 µM IPTG, and the culture was incubated for 4 hours at 18 °C. After harvesting the cells (10 min at 13,000×g; 4 °C), the pellets were resuspended in NZY Bacterial Cell Lysis Buffer (NZYTech) supplemented with 2 mg/mL chicken egg white lysozyme

(Sigma-Aldrich), 50 U/mL GENIUS™Nuclease (Acrobiosystems), and 5 mM MgCl$_2$, and incubated on ice for 30 minutes. The lysate was centrifuged at 13,000 × g for 30 minutes at 4 °C, and the supernatant containing the soluble protein fraction was collected. To this fraction, 4 × Laemmli buffer was added to a final concentration of 1 ×, followed by protein denaturation at 95 °C for 5 minutes. For sodium dodecyl sulfate-polyacrylamide gel electrophoresis (SDS-PAGE) analysis, a 10% polyacrylamide gel was prepared, and electrophoresis was carried out at 180 V for 40 minutes. Soluble proteins were then transferred to an Immobilon-P polyvinylidene fluoride membrane (Merck) using a Transblot SD cell (Bio-Rad) for western blot analysis. The membrane was blocked for 1h at room temperature with blocking solution [5% (w/v) skim milk, 25 mM Tris-HCl pH 7.5, 150 mM NaCl, and 0.1% (v/v) Tween-20]. Detection of ACE2 was performed using an anti-ACE2 primary antibody (1:2,000 dilution, AAF933 from R&D Systems) and HRP-conjugated anti-goat secondary antibody (1:20,000 dilution, sc-2020 from Santa Cruz Biotechnology). Both antibodies were diluted in blocking solution and incubated sequentially for 1h each at room temperature. After ACE2 detection with the ECL™ Western Blotting Detection Reagents kit (Amersham™) the membranes were stripped with 200 mM NaOH and stained with Coomassie Brilliant Blue R-250 [9].

## ACE2 production and purification using pET28g-Gb1_ACE2

To express Gb1-tagged ACE2, conditions were tested, varying expression media [LB, Terrific Broth (TB), LB auto-induction medium (LB AIM), and Terrific Broth auto-induction medium (TB AIM), all purchased from Grisp] and expression duration (4 or 16h). LB AIM and TB AIM are modified versions of LB and TB, respectively, designed for efficient protein expression without the need for IPTG. These auto-induction media contain glucose, which supports initial bacterial growth while repressing protein expression. As glucose is depleted, the cells switch to metabolizing alpha-lactose, which induces protein expression through lac-based promoters, ensuring high protein yields without adding expression inducers, such as, e.g., IPTG. Cell pellets harvested by centrifugation were lysed, and the total soluble protein extracts were obtained by centrifugation, as described above. Protein expression was evaluated by SDS-PAGE and western blot as described above, but proteins were detected with different antibodies. For the expression tests, the His-tagged ACE2 was detected using a polyhistidine antibody (SigmaAldrich, H1029, 1:6000 dilution) and an anti-mouse IgG (Fab specific)-peroxidase antibody (SigmaAldrich, A9917, 1:4000 dilution).

Once the most appropriate expression condition was selected, an expression scale-up was set up. *E. coli* BL21 (DE3) was transformed with pET28g-Gb1_ACE2 as described above, and a single colony was used to inoculate a 30-mL starter culture in LB with 50 µg/mL kanamycin, grown overnight at 37 °C, 140 rpm in a shaking incubator (NB-205LF, N-BIOTEK). Large-scale cultures were prepared in 3-L baffled conical shake flasks, each containing 1 L TB AIM with 50 µg/mL kanamycin, which were inoculated with 10 mL of the overnight starter culture. Cells were grown at 37 °C, 140 rpm, until the OD$_{600}$ reached 1.8–2.0. Then, bacterial cultures were transferred to 18 °C and incubated overnight at 140 rpm. The bacterial cells were harvested by centrifugation at 10,000 × g at 4 °C for 7 min.

For the purification of ACE2, cell pellets were resuspended in phosphate buffer saline (PBS), pH 7.4, supplemented with 1 mg/mL lysozyme (Sigma-Aldrich), 1:5,000 benzonase (Acrobiosystems), 2 mM MgCl$_2$, 200 µM TCEP, 1:500 cOmplete™ EDTA-free protease inhibitor cocktail (Roche) and lysed by sonication with a UP200S Ultrasonic Processor (Hielscher): eight cycles with 50% amplitude and 0.6 cycle for 30 s of sonication and 30 s of incubation on ice. Cell debris were removed by centrifugation at 45,000 × g, at 4 °C for 30 min. The collected supernatant was filtered with a 0.45 µm pore size filter and loaded onto a HisTrap column (Cytiva) previously equilibrated with binding buffer (PBS pH 7.4, 20 mM imidazole), in an Åkta GO (Cytiva) chromatography system. The target protein was eluted with a linear gradient of 500 mM imidazole in PBS pH 7.4. His-Gb1-ACE2 eluted at approximately 240 mM imidazole, and the corresponding fractions were identified by SDS-PAGE. The pooled fractions were then concentrated to approximately 1 mg/mL and buffer-exchanged to PBS pH 7.4 using an Amicon™ Ultra Centrifugal Filter Unit (Merck) with a 50-kDa cut-off. The concentrated protein was incubated overnight with 1:20 (w/w) TEV protease for tag cleavage. The digested protein was then applied to a His Grav-iTrap column pre-equilibrated with PBS pH 7.4 to capture the His-tag and TEV protease, allowing the untagged ACE2 to

elute in the flowthrough. The collected untagged ACE2 was further purified by size-exclusion chromatography on a HiLoad 16/600 Superdex™ 200 pg column (Cytiva) pre-equilibrated with PBS pH 7.4. Peak fractions were analyzed by SDS-PAGE, pooled, concentrated, and stored at −80 °C.

### Analysis of ACE2 by far-UV circular dichroism

To study the secondary structure content of ACE2 expressed in *E.* coli, far-UV circular dichroism (CD) spectra were recorded in a J-815 spectropolarimeter (JASCO) equipped with a CDF-426S Peltier temperature controller (JASCO). The protein sample was diluted to 0.2 mg/mL in PBS pH 7.4 and transferred to a 0.1 cm path quartz cuvette. The spectrum was acquired at 25°C, resulting from four accumulations under the following conditions: scanning speed 50 nm/min; data pitch 0.2 nm; data integration time 2 s; bandwidth 2 nm; and $N_2$ flow 8 L/min. The spectrum of PBS pH 7.4 was collected under the same conditions and subtracted from the ACE2 spectrum.

### Measurement of ACE2 activity

The ACE2 activity was evaluated using the fluorogenic substrate Mca-APK(Dnp) (Enzo Life Sciences) as previously described [10]. Briefly, the proteolytic activity was assayed in 100 mM MES-KOH pH 6.5, 300 mM NaCl, 10 mM $ZnCl_2$, 50 µM Mca-APK(Dnp), 50 nM ACE2, 0.01% Brij-35, and 0.35% dimethyl sulfoxide. Negative controls were prepared where the protein or substrate were replaced by an equivalent volume of PBS buffer or dimethyl sulfoxide, respectively. Substrate hydrolysis was monitored for 60 min by measuring the fluorescence increase (excitation wavelength 328 nm, emission wavelength 393 nm) in a Cary Eclipse plate reader (Varian). The rate of substrate hydrolysis, given in arbitrary units (a.u.) per minute, was calculated over the 15–40 min time course. The statistical analyses (Normality, ANOVA, and Tukey's multiple comparison tests) were performed using Prism 5 for Windows v. 5.00 (GraphPad), and the differences were considered statistically significant if $p < 0.05$.

## Results and discussion

Here, we report the generation of pET28g, an easily customizable plasmid for recombinant protein expression in *Escherichia coli*, and show its applicability for the production of recombinant truncated human ACE2 protein.

The pET28g plasmid was built from pET28a by cloning a Golden Gate cassette containing the *lacZα* reporter gene. The assembly of the final pET28g plasmids is based on Golden Gate cloning and was designed as an extension of the MoClo Toolkit (Addgene, kit #1000000044) [6,8], which makes pET28g a level 1 destination plasmid. pET28g expands the existing pET system and can potentially replace any pET plasmids. If a comprehensive collection of level 0 modules is developed, constructs relying on the backbone of the pET system can be rapidly prepared to test different tags for improving the solubility, detection, and purification of proteins; modulating protein subcellular localization; introducing protease sites for removing protein extensions; and co-expressing two or more proteins. All this can be achieved by cloning the gene of interest into only one level 0 module and combining it, in a single tube, with the pET28g and other selected level 0 modules.

A platform to generate multiple constructs to test protein expression is important, as it is still a significant challenge to predict whether a protein of interest will be soluble when expressed in *E. coli*, and which combination of tags can enhance protein expression and solubility. As reviewed by Peti and Page [11], a comprehensive collection of plasmids in which the gene of interest can be inserted using the same cloning strategy is fundamental to accelerating the screening process. However, classical restriction enzyme-assisted cloning is very time-consuming, as it requires both the plasmids and the inserts to be digested and purified in advance. Plus, it typically relies on the use of multiple restriction enzymes that may be difficult to combine and may demand the preparation of distinct gene inserts flanked by the proper restriction sites. Golden Gate cloning technology significantly reduces the burden of DNA purification, as digestion and ligation steps happen simultaneously in one single tube, without the need to purify the fragments. Also, in contrast with the conventional

restriction enzyme-assisted cloning, a single restriction enzyme is used to assemble a myriad of constructs. Therefore, the pET28g, coupled with the Golden Gate cloning, facilitates the creation of multiple constructs requiring only one initial cloning of the gene of interest.

Kits to assemble transcription units for protein expression in *E. coli* using Golden Gate cloning are already available at Addgene (Kits #1000000080, #1000000059, and #1000000134), but none of them use the parts of the MoClo Toolkit (Kit #1000000044). The tools presented in this work constitute an extension for MoClo Toolkit users. Additionally, our system is based on a well-known expression plasmid, pET28a, which has been widely used and characterized. Notably, in 2020, Shilling et al. [12] reported a 33- to 121-fold increase in protein production yield after restoring the full T7 promoter sequence and modifying the translation initiation region in pET28a. Using the well-known backbone allows any design flaws identified by the scientific community to be fixed at the level 1 plasmid and propagated to the multiple final constructs generated with it. Furthermore, the same Golden Gate cassette can be introduced into other widely used plasmids that use different promoters, allowing for tighter regulation of protein expression, such as pBAD (Thermo Scientific). Constructing different level 1 plasmids enables the use of the developed level 0 modules while taking advantage of other well-known expression systems.

Fig 1 schematizes the cloning strategy used to generate the pET28g plasmid. The Golden Gate cassette, containing a *lacZα* reporter gene flanked by BsaI recognition sites that expose specific fusion sites (Fig 1A), was amplified from a MoClo Toolkit (Addgene, kit #1000000044) level 1 acceptor plasmid. Primers for this amplification were designed to allow the restriction cloning of the PCR product into pET28a using the enzymes NcoI and XhoI (Fig 1B). As depicted in Fig 1C, pET28g retains the backbone of pET28a, namely the T7 promoter, ribosome-binding site (RBS), and T7 terminator, and accommodates a Golden Gate cassette immediately after the RBS.

To use pET28g, new level 0 modules must be created to complete transcription units with the genes of interest, flanked by the tags and signal peptides that best suit tailored protein expression strategies. Modules can be created using the universal level 0 acceptor plasmid or the level 0 acceptors provided with predefined fusion sites within the MoClo Toolkit (Addgene, kit # 1000000044). To systematize the use of the latter, we have planned the pET28g modules using the predefined fusion sites available in the MoClo Toolkit (Addgene, kit # 1000000044 and numbered them from 1 to 7, as detailed in Table 2.

The strategy to prepare level 0 modules depends on the type of acceptor plasmid used, the size of the insert, and the availability of the DNA-fragment of interest (FOI) for amplification. Generally, three distinct strategies can be used to obtain the FOI and insert it into the level 0 acceptor: (i) synthesize the FOI; (ii) design primers to amplify the FOI; (iii) anneal complementary oligonucleotides if the FOI size is below 100 bp. Designing primers to amplify the FOI tends to be less expensive, as it generally requires only the acquisition of the primer pair, but when considering the human resource cost, the overall expense of this strategy can increase significantly. Additionally, it requires the template to be available and free of recognition sites for the type IIS enzymes used in Golden Gate assembly (BsaI and BpiI). If the template is unavailable or requires extensive domestication to remove type IIS enzyme restriction sites, then synthesizing the FOI simplifies the preparation of the level 0 module, reducing the burden on human resources. Nowadays, as this has become a highly cost-effective service, the total costs of this approach—including both service expenses and human resources—may be comparable to or even lower than the primer-based amplification approach. Amplifying or synthesizing FOIs smaller than 100 bp, such as small tags or protease cleavage sites, can be technically challenging. For those specific situations, we suggest using a strategy based on annealing oligonucleotides to create a double-stranded fragment flanked by the fusion site overhangs.

Regardless of the strategy pursued to obtain the FOI, adding the proper adapters to allow Golden Gate assembly into the acceptor plasmid is always required. Figs 2 and 3 schematize the distinct approaches to obtain the FOI reviewed above. The adapters to clone the FOI into the acceptor plasmid depend on whether the universal acceptor or a plasmid with predefined fusion sites is used to prepare the level 0 modules. Therefore, Fig 2 depicts the cloning

strategy when using a universal acceptor plasmid, while Fig 3 depicts the strategy to use plasmids with predefined fusion sites.

For the preparation of level 0 modules using the universal acceptor (Fig 2), two distinct sets of fusion sites are used. A pair of universal fusion sites (highlighted in green in Fig 2) is used for the level 0 assembly upon digestion with BpiI enzyme. A second pair of fusion sites (highlighted in pink in Fig 2) is determined by the position of the level 0 in the final construct (as depicted in Table 2), and must be included in the FOI insert as illustrated in Fig 2B and 2C. This second pair of fusion sites (highlighted in pink in Fig 2) integrates the final plasmid (Fig 2C) in a position that allows for their exposure through the restriction with the BsaI enzyme. It is worth noting that the universal pair of fusion sites (5'-CTCA and 5'-CGAG, highlighted in green in Fig 2) reconstitutes the BsaI recognition site (5′-GGTCTC) in the final plasmid, which is used to expose the fusion sites in the level 1 assembly.

To prepare level 0 modules using an acceptor plasmid with predefined fusion sites (Fig 3), only one pair of fusion sites (highlighted in pink in Fig 3) is necessary to insert the FOI into the level 0 acceptor plasmid. This pair of fusion sites is also used for the level 1 assembly and is chosen according to the position that the module will occupy in the final construct. The FOI must be prepared to expose the fusion sites used to ligate the insert into the acceptor plasmid upon digestion with the BpiI enzyme (Fig 3B). Due to the positioning of the BsaI enzyme recognition sites (5′-GGTCTC) in the acceptor plasmid (highlighted in yellow in Fig 3), the same pair of fusion sites is exposed by BsaI during the level 1 assembly (Fig 3C). S1 Table provides a list of the acceptor plasmids with the predefined fusion sites available in the MoClo Toolkit (Addgene, kit # 1000000044) following the nomenclature proposed here in Table 2.

For standardization purposes, we suggest a nomenclature for all level 0 modules. The name "*p028g*" should be used to identify any level 0 plasmid in the pET28g system. This label should then be followed by the numbers of the fusion sites (see Table 2) used to flank the FOI on the 5' and 3' sides, respectively, allowing the immediate identification of the sites used for FOI assembly. Finally, a short tag should be added to identify the FOI cloned in the module. For example, a module containing an MBP tag flanked by the fusion sites "CCAT" and "AGGT", fusion sites 1 and 3, should be named "p028g13_MBP".

To test this new tool, we prepared level 0 modules to introduce a tag at the N-terminus of a truncated version of the human ACE2 protein, including a cleavage site for TEV protease. To maximize the potential of obtaining soluble protein and preventing its accumulation in inclusion bodies, the MBP and Gb1 tags were selected as they have been shown to enhance translational efficiency and improve the solubility of the desired proteins [13–17]. Another important characteristic of these two tags is that they are not found among the commercial pET plasmids (Novagen). Fig 4 shows the protein expression results for the same protein produced in *E. coli* BL21 (DE3) using three different N-terminal tags: 6×His (from pET28a), 8×His fused with the Gb1 polypeptide (Gb1), and MBP (the last two cloned using pET28g). Based on the primary sequence, the expected molecular weight of each expressed protein is 72 kDa for 6×His, 78 kDa for Gb1, and 112 kDa for MBP. Detection of ACE2 by Western blot (Fig 4A and 4B) shows that the protein was produced with all three constructs, but the amounts of soluble ACE2 produced were not sufficient to be observed in Coomassie-stained membranes (Fig 4C and 4D), which is not unusual for eukaryotic proteins produced in *E. coli* [11]. Despite the low expression levels, we screened expression conditions (S1 Fig) for Gb1-ACE2 because the initial test of the different constructs (Fig 4) suggested slightly better protein production levels with this tag. Then, Gb1-tagged ACE2 production was scaled up in *E. coli* BL21 (DE3) grown in TB AIM (with protein expression induced for 16 h at 18 °C). The Gb1-tagged protein was purified by immobilized metal affinity chromatography (IMAC) and successfully processed using TEV protease. The untagged protein, with a molecular size of 69 kDa, was recovered by reverse IMAC and further polished by size exclusion chromatography (Fig 5), yielding 0.7 mg/L of culture with high purity (approximately 98%), as determined by ImageJ analysis.

To assess the structural integrity and functionality of ACE2 expressed in *E. coli*, the purified protein was analyzed by far-UV circular dichroism (CD) and steady-state kinetic measurements (Fig 6). Far-UV CD analysis of the secondary structure of the recombinantly produced ACE2 revealed a predominantly α-helical profile (Fig 6A), consistent with the

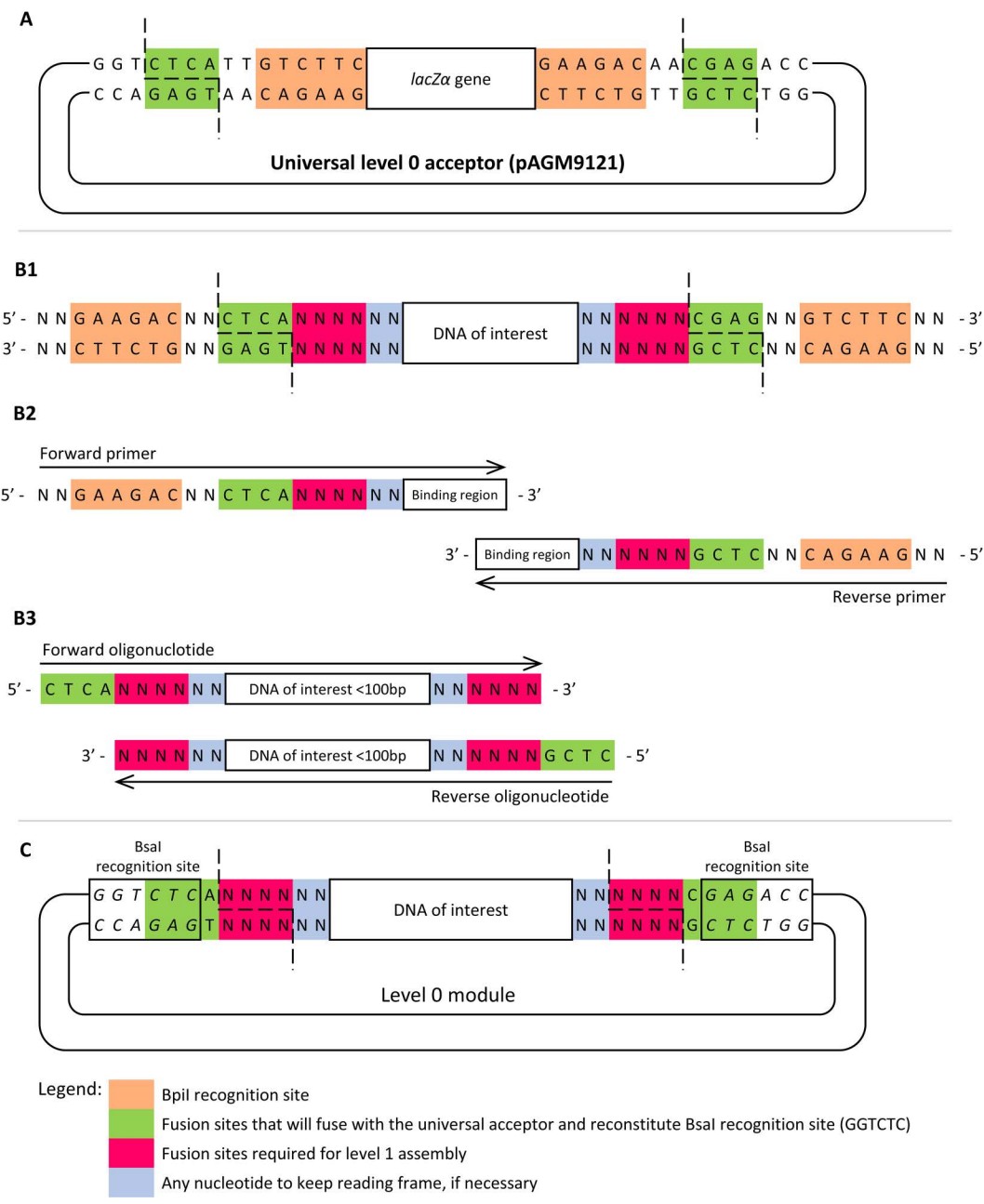

**Fig 2. Strategy to clone level 0 modules using the MoClo Toolkit (Addgene, kit #1000000044) universal level 0 acceptor plasmid (pAGM9121).** **(A)** Schematic representation of the level 0 acceptor plasmid and the cleavage site of the BpiI enzyme; **(B)** Schematic representations of different approaches to prepare the FOI for cloning: (B1) Synthesize the DNA sequence of interest; (B2) Design primers to amplify the DNA of interest; (B3) If DNA size < 100 bp, design oligonucleotides complementary for the DNA of interest region, flanked by uncomplemented fusion sites; **(C)** Schematic representation of the final plasmid obtained upon ligation of the FOI (B) to the universal acceptor plasmid (A) using BpiI for Golden Gate assembly. The BsaI sites reconstituted in the final plasmid are italicized.

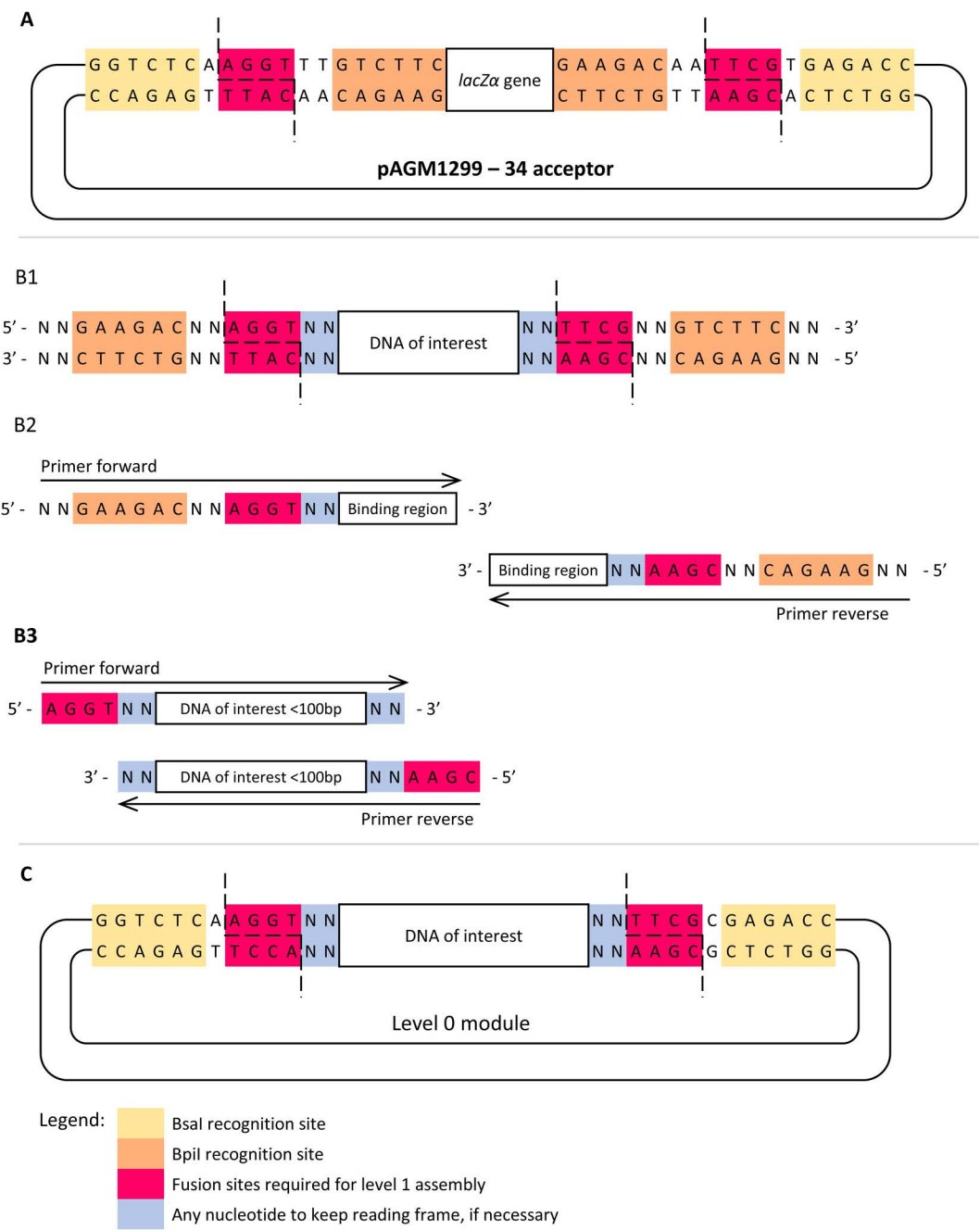

**Fig 3. Strategy to clone level 0 modules using a MoClo Toolkit (Addgene, kit #1000000044) plasmid with the predefined fusion sites (pAGM1299 with fusion sites 3 and 4 was used as an example). (A)** Schematic representation of the level 0 acceptor and the cleavage site of BpiI and BsaI enzymes; **(B)** Schematic representations of different approaches to preparing the FOI for cloning: (B1) Synthesize the DNA sequence of interest; (B2) Design primers to amplify the DNA of interest; (B3) If DNA size < 100 bp, design primers to amplify the DNA of interest,design oligonucleotides complementary for the DNA of interest region, flanked by uncomplemented fusion sites; **(C)** Schematic representation of the final vector obtained upon ligation of the FOI (B) to the universal acceptor vector (A) using BpiI for Golden Gate assembly.

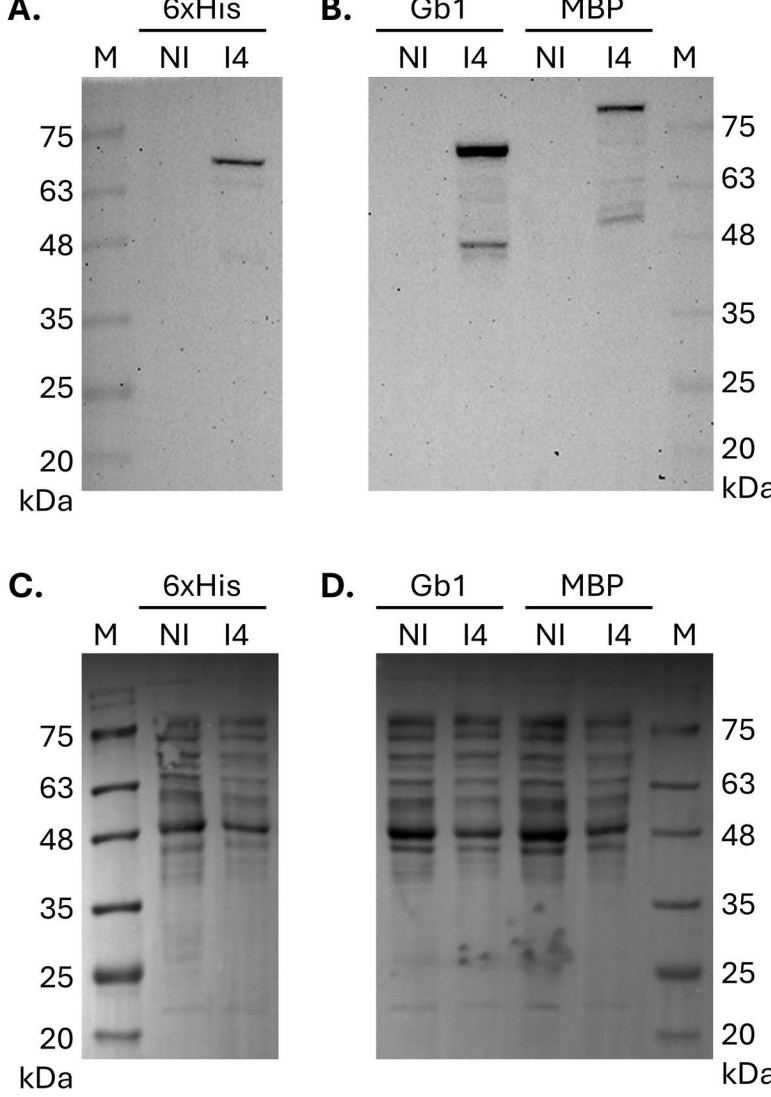

**Fig 4. Detection of ACE2 expressed in *E. coli* BL21 (DE3) using different constructs.** Protein expression as detected in total soluble protein extracts by western blot with anti-ACE2 antibody (A and B) and Coomassie Brilliant Blue staining **(C and D)**. A and C, soluble protein extracts of bacteria transformed with pET28a plasmid expressing ACE2 with a 6x-His tag at the N-terminus; B and D, soluble protein extracts of bacteria transformed with pET28g plasmids expressing ACE2 with an 8×His-Gb1 tag (Gb1) or an MBP tag (MBP) at the N-terminus. NI – Soluble protein extract from bacteria before protein expression induction; I4 – Soluble protein extract from bacteria four hours after protein expression induction; M – Molecular weight marker (MB09002, NZYTech).

three-dimensional structure of the native ACE2 protein (PDB code: 1R42). This suggests that, despite being expressed in a prokaryotic system, the protein retains proper folding and structural integrity. Additionally, we tested the ability of the purified protein to hydrolyze the substrate, Mca-APK(Dnp) (Fig 6B). We were able to show that the ACE2 produced in *E. coli* can cleave the fluorogenic substrate Mca, releasing it from the Dnp quenching group.

To our knowledge, all truncated ACE2 proteins produced in *E. coli* have been purified through the refolding of proteins from inclusion bodies. In some instances, these were expressed as fusion proteins [18,19]. Thus, the ACE2 produced using the expression platform developed in this work represents the first report of this protein being expressed and

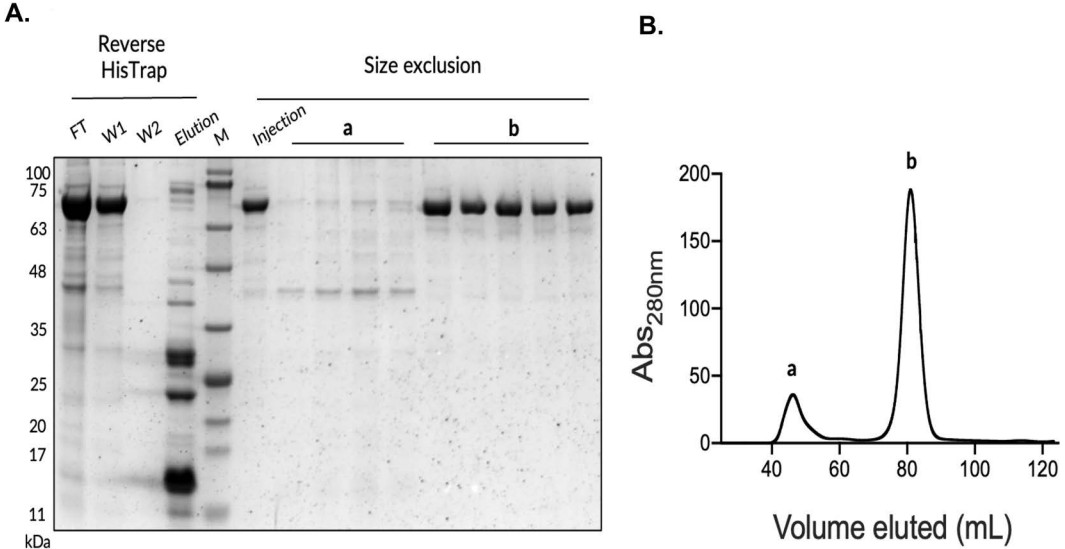

**Fig 5. Purification of ACE2 expressed in *E. coli* BL21 (DE3) using a pET28g plasmid with an 8 × His-Gb1 tag. A)** SDS-PAGE analysis of fractions from the reverse HisTrap purification step, performed after TEV protease cleavage to remove the 8 × His-GB1 tag and His-tagged TEV protease. The flowthrough (FT) contains the cleaved (untagged) ACE2, while the elution fraction contains the His-tagged TEV protease and the cleaved His-tag. Washes 1 and 2 (W1, W2) with PBS pH 7.4 ensured complete recovery of untagged ACE2. The figure also shows fractions (a) and (b) collected from size-exclusion chromatography. **(B)** Size-exclusion chromatography profile of untagged ACE2 purified using a HiLoad 16/600 Superdex™ 200 pg column, where peak (b) corresponds to the untagged ACE2 protein. M – molecular weight marker (MB09002, NZYTech).

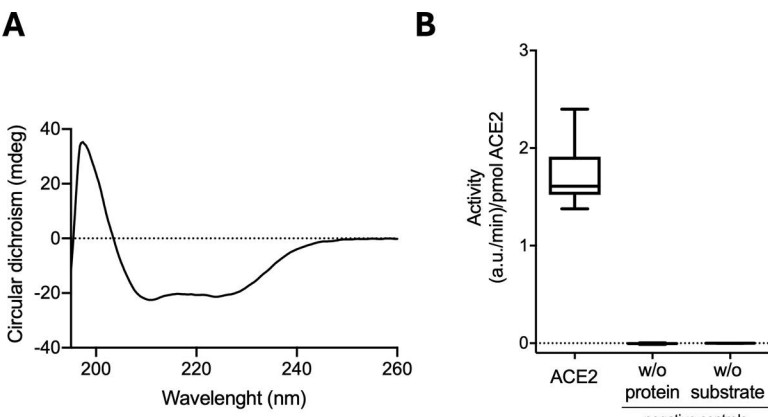

**Fig 6. Assessment of the secondary structure content and enzymatic activity of recombinantly expressed ACE2 in *E. coli*. (A)** Far-UV CD spectra of ACE2 with a characteristic α-helical profile, indicating a predominantly helical secondary structure. Measurements were performed at a protein concentration of 0.2 mg/mL in PBS, pH 7.4, using a 0.1 cm path length cuvette at 25°C. Baseline correction was applied using the buffer CD spectrum. **(B)** Rates of Mca-APK(Dnp) substrate cleavage by recombinant ACE2. The rates are given in fluorescence arbitrary units (a.u.) because the unquenched Mca substrate was not available to prepare a calibration curve. For comparison, the rates observed for the control reactions, in the absence of substrate or protein, were plotted. Differences between the rate determined for ACE2 and the controls are statistically significant.

purified from *E. coli* soluble protein extracts. Moreover, the few reports found in the literature describe the production of different truncated versions of the ACE2 protein, which are not necessarily comparable to the ACE2 version produced here, as different primary sequences can affect protein solubility. Finally, the characteristic α-helical content of the produced protein, together with its ability to hydrolyze Mca-APK(Dnp), indicate a stable and functional conformation. This

implies that the expression pET28g plasmid used here effectively promotes the production of a correctly folded ACE2 protein in *E. coli*, suitable for downstream applications.

## Conclusions

We have successfully created a powerful and versatile tool for recombinant protein expression, by inserting a Golden Gate cassette into the backbone of pET28a, merging the well-recognized properties of the pET system, known for efficiently expressing recombinant proteins, with the versatility of DNA assembly achieved through Golden Gate cloning technology. pET28g and the level 0 modules developed in this work are tools that extend the existing pET collection. Additionally, expanding the tools described here will facilitate the establishment of pipelines to prepare and test multiple constructs, optimizing conditions for achieving high levels of soluble protein expression. The biggest advantage of our system lies in the reliability of the pET backbone, which offers robust, state-of-the-art protein expression.

As a proof of concept, we have produced and purified the peptidase domain of human ACE2 using our plasmid. In fact, this constitutes the first report of successful purification of a correctly folded and active proteolytic domain of human ACE2 protein from the *E. coli* soluble protein fraction.

## Supporting information

**S1 Fig. Analysis of soluble ACE2 expression in *E. coli* BL21 (DE3) under different growth and induction conditions.** (A) 10% SDS-PAGE analysis of soluble protein fractions, prepared as described in the Materials and Methods subsection *ACE2 production and purification using pET28g-Gb1_ACE2*. Samples were induced with IPTG and subjected to a temperature shift from 37 °C to 18 °C for cultures grown in LB and TB media, while cultures in LB AIM and TB AIM were induced only by the temperature shift from 37 °C to 18 °C. Protein expression was analyzed at the indicated time points (0, 4 h, 16h). (B) Western blot analysis of the various conditions using a polyhistidine antibody (Sigma-Aldrich, H1029, 1:3000 dilution). A distinct band at approximately 69 kDa confirms ACE2 expression in the soluble fractions. TB AIM with induction at 18 °C for 16 h shows the most intense and well-defined band at ~ 75 kDa, indicating the highest soluble yield of soluble ACE2. The other conditions show weaker signals or higher background. As these analyses focus on soluble fractions, inclusion bodies are not visible on these gels. Based on these results, TB AIM media with induction at 18 °C for 16 h was selected for the for scale-up production of ACE2. M, molecular weight marker.
(TIF)

**S1 Table. List of acceptor plasmids with predefined fusion sites available in the MoClo Toolkit (Addgene, kit #1000000044).**
(DOCX)

**S1 Protocol. Step-by-step protocol to prepare the level 0 modules for Golden Gate assembly in pET28g, as deposited in protocols.io.** The protocol is deposited under the https://doi.org/10.17504/protocols.io.j8nlk9e75v5r/v1 and can be accessed through the link: https://www.protocols.io/view/preparation-of-level-0-modules-for-golden-gate-ass-j8nlk9e75v5r/v1.
(PDF)

## Author contributions

**Conceptualization:** Inês M. Luís, Isabel A. Abreu.

**Investigation:** Inês M. Luís, Mariana Parada.

**Methodology:** Inês M. Luís, Isabel A. Abreu.

**Resources:** João B. Vicente, Isabel A. Abreu.

**Supervision:** João B. Vicente, Isabel A. Abreu.

**Writing – original draft:** Inês M. Luís, Isabel A. Abreu.

**Writing – review & editing:** Inês M. Luís, Mariana Parada, João B. Vicente, Isabel A. Abreu.

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
