## [Decision Letter · Decision Letter 0]

Dear Dr. Abreu,

Thank you for submitting your manuscript to PLOS ONE. After careful consideration, we feel that it has merit but does not fully meet PLOS ONE’s publication criteria as it currently stands. Therefore, we invite you to submit a revised version of the manuscript that addresses the points raised during the review process.

**ACADEMIC EDITOR**

We look forward to receiving your revised manuscript.

Kind regards,

Hari S. Misra, Ph.D.

Academic Editor

PLOS ONE

Journal Requirements:

We acknowledge Fundação para a Ciência e a Tecnologia (Portugal) for Fellowship for IML (PD/BD/113982/2015) and grants FilliGrain-Protect (DOI:10.54499/PTDC/ASP-PLA/1920/2021), Green-it Base (DOI:10.54499/UIDB/04551/2020) and Programmatic (DOI 10.54499/UIDP/04551/2020) Funding, LS4FUTURE Associated Laboratory (DOI 10.54499/LA/P/0087/2020), MostMicro Programmatic Funding (UIDP/04612/2020), and iNOVA4Health R&D Unit (UIDB/04462/2020, UIDP/04462/2020). We acknowledge 'la Caixa' Foundation for grant HR22-00722/2022.

Additional Editor Comments :

The manuscript has been reviewed by 3 subject experts. They have different comments and a list of suggestions for authors to work on. Along with the reviewers comments, authors should also verify that the recombinant protein made through this technology has functional conformation. Therefore, purification of the recombinant protein and its structural characterization would be needed for this work to become meaningful.

Reviewers' comments:

Reviewer's Responses to Questions

**Comments to the Author**



Reviewer #1: Yes

Reviewer #2: No

Reviewer #3: Yes

2. Has the protocol been described in sufficient detail?

To answer this question, please click the link to protocols.io in the Materials and Methods section of the manuscript (if a link has been provided) or consult the step-by-step protocol in the Supporting Information files.

Reviewer #1: Yes

Reviewer #2: Partly

Reviewer #3: Yes

3. Does the protocol describe a validated method?

Reviewer #1: Yes

Reviewer #2: Yes

Reviewer #3: Yes

4. If the manuscript contains new data, have the authors made this data fully available?

Reviewer #1: Yes

Reviewer #2: N/A

Reviewer #3: Yes

**5. Is the article presented in an intelligible fashion and written in standard English?**

Reviewer #1: Yes

Reviewer #2: Yes

Reviewer #3: Yes

Reviewer #1: The manuscript submitted by Luís and coworkers introduces pET28g, a novel plasmid tool that merges the established robustness of the pET expression system in E. coli with the modular flexibility of Golden Gate cloning. This combination enables efficient screening of constructs to optimize heterologous protein expression. As such, the manuscript presents an innovative approach and should be published. It offers a valuable resource for researchers seeking to optimize bacterial gene expression.

The authors have commendably made a detailed cloning protocol for the pET28g system publicly accessible on protocols.io. Furthermore, they demonstrate the utility of their system by providing a comprehensive example using a human protein known for its challenging solubility in bacterial expression.

I have one major comment and two minor comments to offer.

Major Comment:

To further substantiate the advantages of pET28g, I would recommend expanding the comparative analysis of protein expression between pET28g and pET28a. Specifically, showcasing examples where the modularity of pET28g significantly improves expression yields would be highly beneficial. While acknowledging the time and labor involved in such experiments, including any relevant data from the authors' lab would provide a more comprehensive understanding of pET28g's capabilities.

Minor Comments:

A citation is missing on page 22, three lines before Table 3.

I encountered difficulty locating the protocol on protocols.io. Providing a direct link would greatly assist readers interested in utilizing this resource.

Reviewer #2: We note that soluble expression of ACE2 in E. coli is inherently challenging due to its multiple N glycosylation sites and three disulfide bonds, which cannot form correctly in the reducing environment of the bacterial cytoplasm. Consequently, the apparent solubility of your MBP-ACE2 fusion likely reflects the solubilizing effect of the maltose binding protein tag rather than proper folding of the ACE2 domain. Furthermore, the manuscript predominantly describes incremental modifications of previously published protocols and relies heavily on standard MoClo kits, which limits its novelty for a new journal publication. I recommend either adopting an expression system that supports post translational modifications (e.g., a eukaryotic host or periplasmic targeting with disulfide bond–promoting conditions) or providing rigorous biophysical and functional validation to demonstrate correct ACE2 folding and activity.

Reviewer #3: The manuscript "A new tool for protein expression in Escherichia coli that combines the pET system

with Golden Gate cloning, the pET28g" describes a modification of the pET28 plasmid in order to adopt it to the MoClo Toolkit. This is a nice work describing the plasmid as well as showing its functioning.

I am a bit unsure if this is really a protocol but still I like the work and fell that it is worth publishing.

Please make sure that the links to protocoll.io work once the article is published. It is not working at the moment. As there are no page and line numbers it is difficult to refer to a certain position .

I am missing several references for example in the Material and Methods section for pICH47761, pAMG9121, pDONOR221 an so on. Please make sure that they are given. Also I don´t think that it is sufficient to write " the MoClo kit " . There are many out there and maybe there is only one which is officially named that way, a reference would help a lot.

Specific points:

At this passage "....4× Laemmli buffer was added to a final concentration of 1×, followed by protein denaturation at 95 °C for 5 minutes. Soluble proteins were transferred to an Immobilon-P PVDF membrane (Merck) using a Transblot SD cell (Bio-Rad) to be analyzed by western blot. The membrane was blocked ...." I think you forgot to mention the PAGE step.

Please add description of SDS PAGE, AIM, TB AIM

Add reference to MoClo Kit in Results and Discussion section

"..all by cloning the gene of interest into only one level 0 module without the need to digest and purify multiple plasmids. " I think this is not correct. Maybe the Cut Ligation makes things more easy but you still need plasmids and they are also digested.

similar a bit later" However, classical restriction enzyme-assisted cloning is very time-consuming, as it is necessary to digest and purify both the plasmids and the inserts." In principle also here plasmids and inserts are digested. I would rather emphasize the efficiency and the easy re-use of existing plasmids.

What is meant with "design flaws identified by the scientific community" ?

Is this correctly written? I don´t get it "In the final vector, the 5’-CTCA and 5’-CGAG sequences will reconstitute the BsaI recognition ....." Please improve writing also in the following part. It is difficult to understand.

" or plasmid will be exposed by BsaI during

the level 1 assembly (Error! Reference source not found.C). Please correct

I am a bit confused because after Tab. 1 there are almost all figure legends. I hope that this becomes more clear if the article is put into its final layout.

I don´t Table 3 and am not sure if it is necessary.

Fig 2 and Fig 3 : Description is very similar. Is it possible to combine them into one figure? (B) Schematic representations of different approaches to prepare the insert for cloning... Is this necessary? This is common to all cloning strategies and not specific for MoClo

Fig 4 is a bit of an overkill. I would prefer if you mentioned it in the text.

You almost never mention that inclusion bodies are a problem and you did not mention your strategy for avoiding inclusion bodies. Maybe add a sentence when you describe ACE2 as you model protein.

Fig S1 how was the extract prepared? Would inclusion bodies be visible here on this gel?

**Do you want your identity to be public for this peer review?** For information about this choice, including consent withdrawal, please see our Privacy Policy

Reviewer #1: No

Reviewer #2: **Yes: ** Hamid Reza Karbalaei-Heidari

Reviewer #3: No

---

## [Author Response · Author response to Decision Letter 1]

11 Jun 2025

Response to Reviewers:

Reviewer #1: The manuscript submitted by Luís and coworkers introduces pET28g, a novel plasmid tool that merges the established robustness of the pET expression system in E. coli with the modular flexibility of Golden Gate cloning. This combination enables efficient screening of constructs to optimize heterologous protein expression. As such, the manuscript presents an innovative approach and should be published. It offers a valuable resource for researchers seeking to optimize bacterial gene expression.

The authors have commendably made a detailed cloning protocol for the pET28g system publicly accessible on protocols.io. Furthermore, they demonstrate the utility of their system by providing a comprehensive example using a human protein known for its challenging solubility in bacterial expression.

I have one major comment and two minor comments to offer.

Major Comment:

To further substantiate the advantages of pET28g, I would recommend expanding the comparative analysis of protein expression between pET28g and pET28a. Specifically, showcasing examples where the modularity of pET28g significantly improves expression yields would be highly beneficial. While acknowledging the time and labor involved in such experiments, including any relevant data from the authors' lab would provide a more comprehensive understanding of pET28g's capabilities.

R: We appreciate the reviewer’s suggestion and fully agree that comparative expression data would further support the utility of pET28g. Notwithstanding, we propose that the use of pET28g increases the chances of producing soluble recombinant in high yields because it allows for modularity in the preparation of constructs using the backbone of the pET expression system, while pET28a only allows the production of recombinant protein with His-tag. Thus, a direct comparison is not the point, as we see it. Rather, this tool greatly facilitates testing multiple tags requiring a much lower investment in terms of resources and time.

In addition, we have already applied pET28g in our laboratory to express other proteins unrelated to ACE2. As an example, a protocol currently under preparation describes the successful expression of gibberellin 3-oxidase (GA3ox) using pET28g. This protocol is available to reviewers via the following private link: https://www.protocols.io/private/5F525F713C8B11F080530A58A9FEAC02.

We hope this clarifies the potential broader use of pET28g.

Minor Comments:

A citation is missing on page 22, three lines before Table 3.

R: The missing citation has been added. In addition, we have thoroughly reviewed all other citations and cross-references throughout the manuscript and made corrections wherever necessary to ensure consistency and accuracy. We thank the reviewer for pointing out our mistake.

I encountered difficulty locating the protocol on protocols.io. Providing a direct link would greatly assist readers interested in utilizing this resource.

R: Thank you for bringing this to our attention. We were under the impression that protocols.io entries accompanying a manuscript should remain in “Reserved DOI” status and be included as supplemental material. Accordingly, we followed this procedure and included in the manuscript the DOI generated during the “Reserve DOI” process.

We now realize that this may have caused access issues for reviewers, which we were previously unaware of, as the link remains inactive until the protocol is formally published. Since published protocols can no longer be edited, we intend to publish the protocol only upon acceptance of the manuscript to allow updating the reference to the manuscript in the protocol. At present, it remains in “Reserved DOI” status with the DOI: [10.17504/protocols.io.j8nlk9e75v5r/v1].

To facilitate future access, the corresponding public link has been added to the manuscript. However, if this link changes upon formal publication of the protocol (which remains unclear to us), we respectfully request that it be updated accordingly.

In the meantime, reviewers may access the protocol directly via the following private link:

https://www.protocols.io/private/892F9651DDA411EFACC60A58A9FEAC02

Reviewer #2: We note that soluble expression of ACE2 in E. coli is inherently challenging due to its multiple N glycosylation sites and three disulfide bonds, which cannot form correctly in the reducing environment of the bacterial cytoplasm. Consequently, the apparent solubility of your MBP-ACE2 fusion likely reflects the solubilizing effect of the maltose binding protein tag rather than proper folding of the ACE2 domain. Furthermore, the manuscript predominantly describes incremental modifications of previously published protocols and relies heavily on standard MoClo kits, which limits its novelty for a new journal publication. I recommend either adopting an expression system that supports post translational modifications (e.g., a eukaryotic host or periplasmic targeting with disulfide bond–promoting conditions) or providing rigorous biophysical and functional validation to demonstrate correct ACE2 folding and activity.

R: In our study, soluble ACE2 was expressed in E. coli in several formats, including a fusion with an N-terminal Gb1 tag. Crucially, the tag was enzymatically removed, yielding tag-free ACE2 protein that remains properly folded, as demonstrated by far-UV circular dichroism (CD) data now included in the revised manuscript in Figure 6, panel A. Furthermore, and to provide functional validation, we performed enzymatic assays showing that the E. coli-produced ACE2 hydrolyzes the fluorogenic substrate Mca-APK(Dnp) (Figure 6, panel B). These results are detailed in the Materials and Methods and Results sections. These additional results conclusively show that functional peptidase domain of human ACE2 can be produced in E. coli, using our system.

As the success of protein production depends greatly on the intrinsic properties of each target protein, we acknowledge that our system will not be universally applicable. Nevertheless, we remain confident in the utility of this new tool and believe that sharing it with the community will be of value.

Reviewer #3: The manuscript "A new tool for protein expression in Escherichia coli that combines the pET system with Golden Gate cloning, the pET28g" describes a modification of the pET28 plasmid in order to adopt it to the MoClo Toolkit. This is a nice work describing the plasmid as well as showing its functioning.

I am a bit unsure if this is really a protocol but still I like the work and fell that it is worth publishing.

Please make sure that the links to protocoll.io work once the article is published. It is not working at the moment. As there are no page and line numbers it is difficult to refer to a certain position.

R: The reviewer is correct as a working link will only be available when the protocols.io protocol is published - please see response to reviewer 1, who raised the same issue, above.

Page and line numbers were added, using PLOS ONE template – we apologize for our omission.

I am missing several references for example in the Material and Methods section for pICH47761, pAMG9121, pDONOR221 an so on. Please make sure that they are given. Also I don´t think that it is sufficient to write " the MoClo kit ". There are many out there and maybe there is only one which is officially named that way, a reference would help a lot.

R: We agree with the reviewer’s assessment. To clarify the source and identity of the plasmids used in this study, we have added the following sentence at the beginning of the Materials and Methods section:

“The MoClo plasmids referenced throughout this work are part of the MoClo Toolkit deposited in Addgene under catalogue number #1000000044 [6,8].”

In addition, small but important edits were made throughout the Materials and Methods and Results and Discussion sections to ensure the origins of all plasmids mentioned—such as pICH47761, pAGM9121, and pDONR221—are clearly stated.

The most significant revision appears in the subsection Cloning of the gene of interest, which now reads:

“The gene of interest used to test this new plasmid encodes a truncated version of the human protein angiotensin-converting enzyme 2 (ACE2, UniProt Acc. No. Q9BYF1). The gene was amplified from a previously cloned pDONR221 plasmid containing the peptidase domain of the ACE2 gene (residues 19 to 615). Briefly, the gene with codon optimization for protein expression in E. coli was synthesized with attB adaptors and cloned into the pDONR221 plasmid (Thermo Fisher Scientific) using Gateway cloning. Then, the peptidase domain of ACE2 was amplified from the pDONR211 cloned with the ACE2 gene using the primers with the following sequences: 5'-TTGAAGACATCTCATTCGACCATTGAAGAACAGGCGA (F1_GG_Ec-hACE2_TTCG) and 5'-TTGAAGACAACTCGAGCGTTAATCTGCATACGGGCTC (R1_GG_Ec-hACE2_CGCT). “

These edits were made to include information on how the pDONR221 containing ACE2 gene was constructed and the origin of the pDONR221 plasmid..

Specific points:

1. At this passage "....4× Laemmli buffer was added to a final concentration of 1×, followed by protein denaturation at 95 °C for 5 minutes. Soluble proteins were transferred to an Immobilon-P PVDF membrane (Merck) using a Transblot SD cell (Bio-Rad) to be analyzed by western blot. The membrane was blocked ...." I think you forgot to mention the PAGE step.

Please add description of SDS PAGE, AIM, TB AIM.

R: We thank the reviewer for their comment. The descriptions were added where appropriate.

The text starting at line 175, now reads:

“The lysate was centrifuged at 13,000 ×g for 30 minutes at 4 °C, and the supernatant containing the soluble protein fraction was collected. To this fraction, 4× Laemmli buffer was added to a final concentration of 1×, followed by protein denaturation at 95 °C for 5 minutes. For sodium dodecyl sulfate-polyacrylamide gel electrophoresis (SDS-PAGE) analysis, a 10% polyacrylamide gel was prepared, and electrophoresis was carried out at 180 V for 40 minutes. Soluble proteins were then transferred to an Immobilon-P polyvinylidene fluoride membrane (Merck) using a Transblot SD cell (Bio-Rad) for western blot analysis.”

The text starting at line 191, now reads:

“To express Gb1-tagged ACE2, conditions were tested, varying expression media [LB, Terrific Broth (TB), LB auto-induction medium (LB AIM), and Terrific Broth auto-induction medium (TB AIM), all purchased from Grisp] and expression duration (4 or 16h). LB AIM and TB AIM are modified versions of LB and TB, respectively, designed for efficient protein expression without the need for IPTG. These auto-induction media contain glucose, which supports initial bacterial growth while repressing protein expression. As glucose is depleted, the cells switch to metabolizing alpha-lactose, which induces protein expression through lac-based promoters, ensuring high protein yields without adding expression inducers, such as e.g., IPTG.”

2. Add reference to MoClo Kit in Results and Discussion section

R: As explained above, in response to the reviewer main comment, a sentence was added in the beginning of the Materials and Methods section, and small edits were made throughout the Materials and Methods, and Results and Discussion sections to clarify the origins of the plasmids mentioned.

3. "..all by cloning the gene of interest into only one level 0 module without the need to digest and purify multiple plasmids." I think this is not correct. Maybe the Cut Ligation makes things more easy but you still need plasmids and they are also digested.

R: The reviewer is correct in pointing out the lack of accuracy in the sentence. Our intention was to emphasize the advantages of Golden Gate assembly over conventional restriction enzyme-based cloning, which typically requires multiple plasmid backbones, various restriction enzymes, and sequential digestion and purification steps to assemble different constructs.

To improve clarity and accuracy, we have revised the sentence, which now reads (lines 265-267):

“All this can be achieved by cloning the gene of interest into only one level 0 module and combining it, in a single tube, with the pET28g and other selected level 0 modules.”

This statement assumes a context in which “a comprehensive collection of level 0 modules is developed” (line 261). In such a scenario, only the gene of interest needs to be cloned, after which different expression constructs can be generated simply by combining it with selected level 0 plasmids, enabling rapid screening under various expression conditions.

4. similar a bit later" However, classical restriction enzyme-assisted cloning is very time-consuming, as it is necessary to digest and purify both the plasmids and the inserts." In principle also here plasmids and inserts are digested. I would rather emphasize the efficiency and the easy re-use of existing plasmids.

R: The paragraph containing that sentence (now starting at line 273) was extensively revised to clarify the differences between conventional cloning and the modular Golden Gate approach. The revised version emphasizes that, with the pET28g system and Golden Gate cloning, a single initial cloning step of the gene of interest into a level 0 module is sufficient to generate multiple expression constructs, allowing for rapid screening of different expression conditions to identify the optimal configuration for protein production. The paragraph now reads:

“However, classical restriction enzyme-assisted cloning is very time-consuming, as it requires both the plasmids and the inserts to be digested and purified in advance. Plus, it typically relies on the use of multiple restriction enzymes that may be difficult to combine and may demand the preparation of distinct gene inserts flanked by the proper restriction sites. Golden Gate cloning technology significantly reduces the burden of DNA purification, as digestion and ligation steps happen simultaneously in one single tube, without the need to purify the fragments. Also, in contrast with the conventional restriction enzyme-assisted cloning, a single restriction enzyme is used to assemble a myriad of constructs. Therefore, the pET28g, coupled with the Golden Gate cloning, facilitates the creation of multiple constructs requiring only one initial cloning of the gene of interest.”

5. What is meant with "design flaws identified by the scientific community"?

R: The phrase refers specifically to findings such as those reported by Shilling et al., who identified flaws in the design of pET28a that can be fixed to improve protein expression. The manuscript was updated to make this clear. At line 286, it now reads:

“Additionally, our system is based on a well-known expression plasmid, pET28a, which has been widely used and characterized. Notably, in 2020, Shilling et al. [12] reported a 33- to 121-fold increase in protein production yield after restoring the full T7 promoter sequence and modifying the translation initiation region in pET28a. Using the well-known backbone allows any design flaws identified by the scientific community to be fixed at the level 1 plasmid and propagated to the multiple final constructs generated with it.”

6. Is this correctly written? I don´t get it "In the final vector, the 5’-CTCA and 5’-CGAG sequences will reconstitute the BsaI recognition ....." Please improve writing also in the following part. It is difficult to understand.

R: Yes, we believe the original sentence is technically correct, but we agree that the explanation may be difficult to follow. To improve clarity the paragraphs describing the level 0 cloning strategies—illustrated in Figures 2 and 3—have been thoroughly rewritten to make the process more accessible to the reader. The two paragraphs starting at line 366, now read:

“For the preparation of level 0 modules using the universal acceptor (Fig 2), two distinct sets of fusion sites are used. A pair of universal fusion sites (highlighted in green in Fig 2) is used for the level 0 assembly upon digestion with BpiI enzyme.

---

## [Editor Report · Decision Letter 1]

pET28g: A Golden Gate-compatible pET vector for protein expression in Escherichia coli, validated by production of functional human ACE2

PONE-D-25-13035R1

Dear Dr. Abreu,

We’re pleased to inform you that your manuscript has been judged scientifically suitable for publication and will be formally accepted for publication once it meets all outstanding technical requirements.

Kind regards,

Hari S. Misra, Ph.D.

Academic Editor

PLOS ONE
---

## [Editor Report · Acceptance letter]

PONE-D-25-13035R1

PLOS ONE

Dear Dr. Abreu,

I'm pleased to inform you that your manuscript has been deemed suitable for publication in PLOS ONE. Congratulations! Your manuscript is now being handed over to our production team.

Kind regards,

on behalf of

Professor Hari S. Misra

Academic Editor

PLOS ONE